# Access to Digital Information and Protective Awareness and Practices towards COVID-19 in Urban Marginalized Communities

**DOI:** 10.3390/healthcare10061097

**Published:** 2022-06-13

**Authors:** Siwarat Pattanasri, Thi Phuoc Lai Nguyen, Thanh Bien Vu, Ekbordin Winijkul, Mokbul Morshed Ahmad

**Affiliations:** 1Department of Public Works and Town & Country Planning (DPT), National and Regional Planning Bureau, Bangkok 10310, Thailand; siwaratpattanasri@gmail.com; 2Department of Development and Sustainability, School of Environment, Resources and Development, Asian Institute of Technology, Pathum Thani 12120, Thailand; morshed@ait.asia; 3Department of Land Management, Faculty of Natural Resources and Environment, Vietnam National University of Agriculture, Hanoi 131000, Vietnam; vtbienqldd@gmail.com; 4Department of Energy, Environment and Climate Changes, School of Environment, Resources and Development, Asian Institute of Technology, Pathum Thani 12120, Thailand; ekbordinw@ait.asia

**Keywords:** digital competence, digital inequality, knowledge, practice, slum communities (Thailand)

## Abstract

Due to digital inequality, poor living, and health care conditions, marginalized people are the most vulnerable group to the COVID-19 pandemic. This study examined how digital information influences knowledge, practices, threat appraisals, and motivation behaviors of urban marginalized communities. It examined slum people’s digital competencies, their access to COVID-19 online information, and their trust in COVID-19 information provided by both online and offline media. A total of 453 slum people in Bangkok city, Thailand were surveyed, and multiple regression was performed to examine whether socio-demographic factors influence the access to online communication of slum people. We hypothesized that access to online information might affect marginalized people’s awareness of COVID-19 and resulted in greater levels of their practices and protective behaviors. The finding showed that slum people who had access to online information tended to have a better awareness of self-protection against COVID-19, while elderly, female, and foreign migrant workers faced a number of constraints in accessing COVID-19 online information. Such results are important considering the pandemic is compelling societies to turn toward digital technologies to confront the COVID-19 pandemic and address pandemic-related issues. We also discuss how to enhance the role of digital communication in helping urban marginalized communities during and after the pandemic.

## 1. Introduction

The coronavirus (COVID-19) has spread to more than 200 countries around the world, causing 2.5 million deaths and 113 million infections during the pandemic [1]. In this greatest challenge of the global health crisis, digital communication becomes an important tool for governments, and public and private organizations to disseminate daily COVID-19 information and protective practice guidelines to improve people’s awareness and behavior-related health protection. Governments and public and private organizations also use digital communication to connect with a community via online channels such as social media, email, etc. [2]. Digital communication can help to increase people’s knowledge and awareness [3,4] and risk perceptions related to infectious disease, disaster, and cybercrime [5,6,7]. During the COVID-19 pandemic, the use of digital activities has increased tremendously, especially in countries with strict lockdown regulations [8]. Great online communication efforts were made by governments and public health organizations across the world to raise communities’ awareness about practicing social distancing and hygiene guidelines to battle the COVID-19 pandemic. For all the progress that has been made in the fight against COVID-19, marginalized communities have been left behind in accessing online information. Many studies on digital inequality suggest that people have different digital competence due to their quality of internet access and skills, which may then influence the knowledge and practices they can reap from communication technologies [9,10]. However, digital inequalities may be further reinforced during the COVID-19 pandemic by the lack of digital support [11]. Furthermore, the challenge of disinformation in COVID-19 in many countries caused citizens to undermine trust, amplify fears, and sometimes lead to harmful behaviors [12].

In this paper, we will address the following questions: What are the levels of digital competence of slum people? What sources of COVID-19 information do they refer to? Which factors influence their access to digital information? Finally, how does digital information affect people’s knowledge, practices, and protective motivation behaviors towards the pandemic? We hypothesized slum people with access to digital information would have better knowledge, practices, and protective motivation behaviors towards the pandemic than those without access to digital information.

## 2. Materials and Methods

### 2.1. Site Selection

Khlong Toei slum communities, Bangkok, Thailand were selected for the case study. Khlong Toei slum communities comprise the largest slum in Thailand with 49,225 households and a population of more than 100,000 [13].

It is a place with a highly dense population and is most vulnerable to COVID-19. There are seasonal workers in Khlong Toei port and it has the largest fresh and retail markets in Bangkok which have been severely impacted during the COVID-19 crisis. Due to many conditions in slum communities including poverty, poor WASH (Water, Sanitation, and Hygiene) and living conditions, this has made it difficult to access digital information and services, and there are no adequate facilities to help them prevent infectious diseases [14].

### 2.2. Research Design

The research used quantitative methods to explore and obtain new information and understand slum communities’ lived experiences and digital communication access during the pandemic. Data collection was carried out by questionnaire surveys in the two subcommunities of Lock 1-2-3 and Ban Guay, with a total population of 9125. Using Slovin’s Formula (1), at 5% significance, the calculated sample size required was 383. To increase the reliability of the data analyses, we oversampled cases to 453. Based on the populations in each site, the questionnaire survey was conducted with 385 people in Lock 1-2-3 sub-communities and 68 people in Ban Guay sub-communities (Figure 1).
n = n = N/(1 + Ne^2^)(1)
where,

n = Sample size.

N = Population size.

e = the significance (5%).

**Figure 1 healthcare-10-01097-f001:**
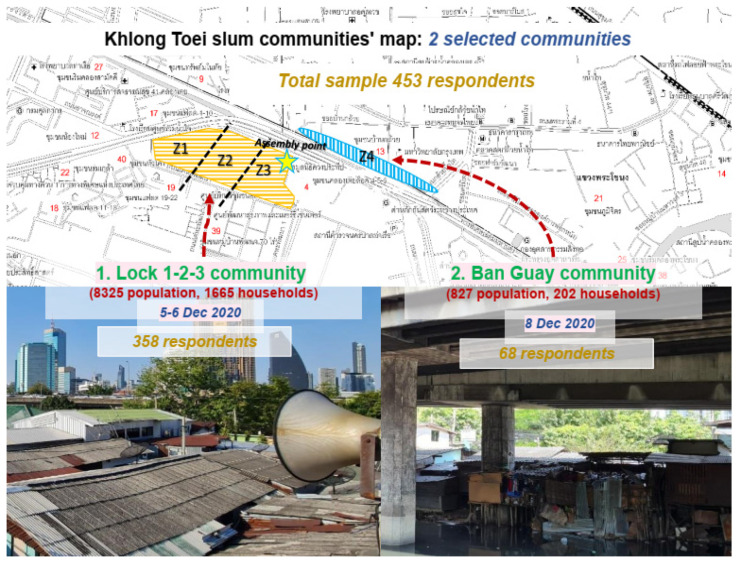
Lock 1-2-3 and Ban Guay, Khlong Toei slum communities, Bangkok, Thailand.

The Snowball sampling method was used as this is suitable for hidden populations, particularly in slum communities which have little information or unknown population behavior and unidentified population census data [15].

Thus, 220 respondents with access to digital information and 233 with non-access to digital information were selected for the survey sample. To reach the coverage of different socio-economic, ethnic, and cultural characteristics of the whole slum community population, some other criteria were set for the sampling including ethnicity, age range, gender, health condition, and disabilities. The population of Khlong Toei slum communities was obtained from the National Statistical Office of Thailand (2019) and the Duang Prateep Foundation.

The questionnaire survey consisted of three parts:(1)Demographic information included age, gender, nationality, occupancy, education, family member, marital status, and state of residential occupancy.(2)Digital competence and skills and information sources included digital device occupancy and usage, digital competence during the pandemic, online service and information evaluation, reliability evaluation of information sources.(3)COVID-19 protection awareness and practices.

The questionnaire survey was conducted using Thai language by eleven trained field assistants with the supervision of three staff from the Duang Prateep Foundation who have extensive experience in working with these slum communities. All field assistants were trained to apply ethical consideration, privacy, and integrity in social surveys.

### 2.3. Data Analysis

Statistical tools were applied for the analysis of questionnaire survey data. Fisher’s exact test was used to compare respondent profiles’ access to digital information among socio-demographic, ethnic, and cultural groups. The Chi-square test was used to compare respondents’ access to digital information and the type of information sources. All statistical analyses were conducted in IBM SPSS Statistics 26 with a significance level of less than 5%.

Multiple regression analysis was used to determine factors influencing the respondents’ levels of marginalized people’s access to digital information, knowledge, practices, and attitudes regarding prevention and protection against COVID-19. The model was developed to explain the relationships between multiple predictor variables including age, gender, nationality, legal marital status, family members, state of residential occupancy, level of education, and occupation.

The multiple regression equation explained above is as follows:Yi = ß0 + ß1X1 + ß2X2 + …… + ß19X19 + ε
where,

Yi = Dependent variable of the regression.

X1, X2, …, X8 = Independent variables of the regression.

ß0 = Intercept term.

ß1, ß2, …, ß8 = Slope coefficients for X1, 2 …, 8.

ε = Error term for the i-th observation.

## 3. Results

### 3.1. Socio-Demographic Characteristics of the Respondents

#### 3.1.1. Marginalized Groups’ Profile

Table 1 shows the demographic profiles of two slum community groups: those with access and those with non-access to digital information. The difference was not statistically significant between the two groups of respondents in terms of gender, family members, and house occupancy. The results revealed that there were differences in terms of ethnicity (*p* < 0.001). Accordingly, Thai people had more access to digital COVID-19 information during the crisis than foreigners. Respondents who can access digital information tended to be middle-aged adults while the non-digital access group tended to be elderly people (*p* < 0.001), meaning that digital literacy was a challenge for elderly people. Low-skilled migrants and people who lost jobs during the crisis were also unable to afford digital devices and internet services. Non-digital access groups have lower education levels (*p* < 0.001), and single status (*p* = 0.001), and most of them are daily wage workers (*p* < 0.001).

#### 3.1.2. Marginalized Groups, Digital Communication, and Information Access

Figure 2 shows that most respondents (59%) used a smartphone to access digital information. However, Figure 3 shows that almost half of respondents were not able to search for COVID-19 information, send or share knowledge or news about COVID-19, create a post on online channels for asking questions with government officers or relevant organizations, buy COVID-19 online services, ask for online help from governmental organizations. These people received daily news and information from offline channels during the pandemic.

Figure 4 shows the percentage of people’s beliefs on the trustworthiness of COVID-19 sources. Slum people believed in the information provided by the TV rather than other sources. They considered online COVID-19 information given by social media as not trustworthy because many of them believed that many organizations tried to create disinformation regarding COVID-19 cases. Thus, they chose to trust community leaders, local healthcare workers, and Duang Prateep Foundation staff (NGO) who work closely with them.

Table 2 shows that there was a significant difference in perceptions of the COVID-19 situation between the two groups. People with digital access highly perceived about ‘COVID situation in Thailand’ (x^2^ = 7.070, df = 1, *p* = 0.008), and the global situation (x^2^ = 4.525, df = 1, *p* = 0.033). People with online access higher perceived the urgent announcements and news from the government (x^2^ = 8.592, df = 1, *p* = 0.003) than non-access to online information groups. Although TV is the main source of information for slum communities, people who have access to online information tended to have a higher perception of the COVID-19 situation in Thailand and the world.

### 3.2. Factors Influencing Slum People’s Access to Digital COVID-19 Information during the Pandemic

Table 3 reports factors affecting slum people’s access to digital information. Age, gender, and ethnicity affected the access of slum people to digital COVID-19 information during the crisis. Younger people directly received more online information than older people (β = 0.610, *p* < 0.001). Men tended to have better access to online information (β = 0.110, *p* = 0.003). Foreign migrants had less access to digital COVID-19 information than Thai migrants (β = −0.169, *p* < 0.001).

## 4. Discussion

### 4.1. Role of Access to Online Information in Urban Marginalized People’s Awareness, Practices, and Protective Motivation Behaviors towards COVID-19

The findings showed that although the majority of urban marginalized people preferred to watch TV for accessing COVID-19 information as they do not trust the online information and/or they do not have access to digital devices and services, people who had more access to online information via social media, internet website, and other platforms tended to have better knowledge and practices for coping with the COVID-19. These findings are aligned with other studies [16,17]. With digital information playing such an important role, the government should provide real-time updates, and accurate information and clarify uncertainties to enhance citizens’ public health awareness during the crisis [18]. Knowledge is an important factor that has a great influence on health behavior [19,20,21] as it shapes the perception of risks which results in people’s severity and vulnerability beliefs [22]. COVID-19 knowledge drives one’s practices resulting in better health protection actions and protective intentions [23,24].

The research findings confirm the hypothesis that digital communication plays a vital role in enhancing the awareness and practices of marginalized people who need speedy information on infectious situations and prevention measures. It emphasized that the pandemic is the major driver that accelerates the role of digital communication in this era [25,26,27]. There is an increasing adoption of digital communication in many sectors to help to boost human activities and the response to COVID-19 [11,28,29]. Social media helps people to have a higher perception of risk, knowledge, and adoption of preventive behaviors towards COVID-19 [30].

### 4.2. Digital Inequality among Urban Marginalized People

Findings showed the digital inequality in slum communities. Half of the surveyed people were not able to reach COVID-19 information and online services. Factors such as age, gender, and nationality affected urban marginalized peoples’ access to online information during the crisis. The results showed elderly people, disabled people, people who occupy unstable jobs, low-education, women, and non-Thai nationals had limits in accessing digital information. Age, language barrier, and poverty are among the barriers to marginalized people from accessing digital information and response capacity [31,32]. Previous studies such as [9,11,33] also highlighted that sociodemographic factor such as gender, race and ethnicity, generation gap, and income cause digital inequality in lower-income and minority groups.

Results also highlighted the difference between poor and rich people in accessing digital information [34]. Poor people mostly faced life-threatening conditions such as unaffordable personal protective equipment, low education, low digital literacy, and inequality in access to the internet which can lead them to have a bias towards protective behavioral intentions [10,30,32,35]. Furthermore, women are restricted to use digital devices in many developing countries due to religious and social norms. The elderly group and migrants are likely to experience disruptions in their online information access because of the high expense of budget devices and internet service plans [36]. Many low-income people struggle to reach information and support from online sources during the crisis [34,37]. Digital inequality access during a pandemic can cause a higher risk for the marginalized groups both in terms of social and health impacts such as non-access or unproper healthcare assistance [11,32].

## 5. Conclusions

Although the study was carried out in a small slum community in Bangkok, this study depicts the role of digital information in enhancing the knowledge and practices that influence the protective behaviors of urban marginalized people from different socio-demographic-nationality backgrounds. Although many efforts from governments and community development organizations in digital communications have been made during the COVID-19 pandemic to enhance citizens’ knowledge, perceived risk, and protective practices, gaps and challenges in digital communication for marginalized people still remain. The study revealed the reality of digital inequality among society and marginalized people. Many vulnerable and marginalized people still have no chance to access digital communication because of their socio-economic status and other personal and social barriers.

The research findings could contribute to communication planning for any future pandemics considering the role of digital equality access among marginalized groups. Governments and organizations should enhance the coverage, efficiency, and effectiveness of digital communication to marginalized people by providing various types of communication tools, media, and formats, resulting in more effective health care for marginalized people in pandemics.

Access to health and disease communication during the COVID-19 pandemic is a basic human right for marginalized people to enhance their well-being and protect themselves against infection risks. The study appeals to the intervention programs of international and local development organizations, governments, and civil societies focusing on education and digital competence pieces of training for marginalized groups, improving healthcare facilities, and digital infrastructure among marginalized communities. Targeting the marginalized people not only helps the vulnerable communities, but also protects the whole community at risk in the pandemic.

## Figures and Tables

**Figure 2 healthcare-10-01097-f002:**
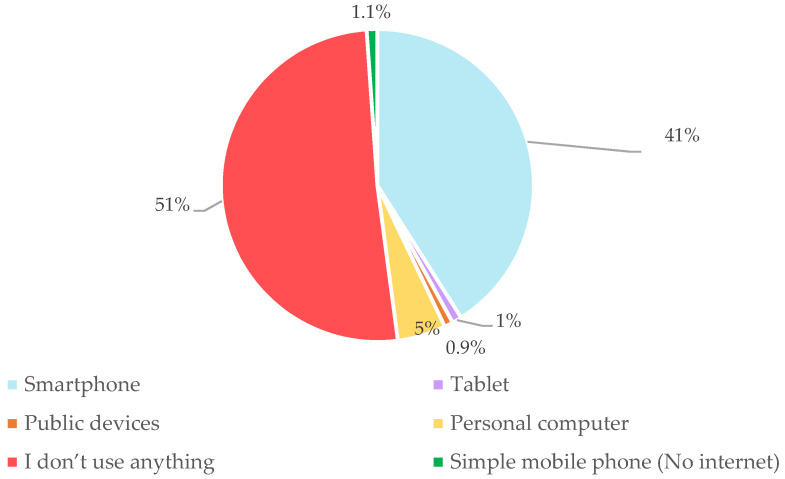
Marginalized people’s digital device usage (*n* = 220).

**Figure 3 healthcare-10-01097-f003:**
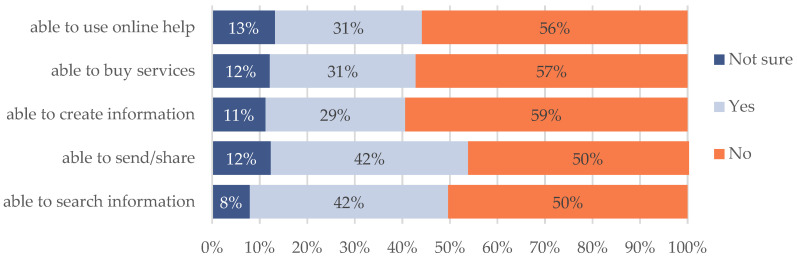
Marginalized people’s digital competence (*n* = 220).

**Figure 4 healthcare-10-01097-f004:**
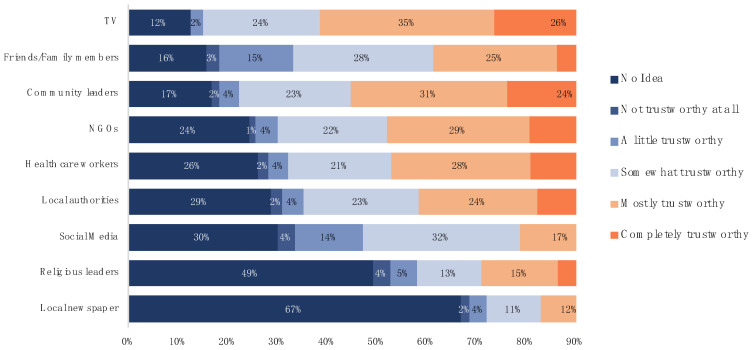
Reliability of COVID-19 information sources. (Five scales: lowest trust (darkest color) to highest trust (lightest color) (*n* = 453).

**Table 1 healthcare-10-01097-t001:** Sample demographics and comparisons among respondents. Comparison between the digital information access and non-access groups (*n* = 453).

Variable	Frequency (Percent)	Access to Digital Communication (Frequency/Percent)	Non-Access Digital Communication (Frequency/Percent)	*p*-Value ^a^
Total number of respondents	453	220	233	
**Age (Years)**				
15–17	42 (9.3%)	-	42 (18.0%)	<0.001 *
18–35	168 (37.1%)	89 (40.5%)	79 (33.9%)	
36–59	136 (30%)	126 (57.3%)	10 (4.3%)	
60–90	107 (23.6%)	5 (2.2%)	102 (43.8%)	
**Gender**				
Male	220 (48.6%)	105 (47.7%)	115 (49.4%)	0.706
Female	232 (51.2%)	114 (51.8%)	118 (50.6%)	
Transgender	1 (0.2%)	1 (0.5%)	-	
**Nationality**				
Thai	401 (88.5%)	220 (100%)	181 (77.7%)	<0.001 *
Myanmar	28 (6.2%)	-	28 (12%)	
Laos	8 (1.8%)	-	8 (3.4%)	
Cambodia	12 (2.6%)	-	12 (5.2%)	
Non-nationality	4 (0.9%)	-	4 (1.7%)	
**Legal marital status**				
Single	205 (45.2%)	86 (39.1%)	119 (51.1%)	0.001 *
Married	195 (43.1%)	112 (50.9%)	83 (35.6%)	
Separated	13 (2.9%)	5 (2.3%)	8 (3.4%)	
Cohabitation	15 (3.3%)	11 (5%)	4 (1.7%)	
Widow(er)	25 (5.5%)	6 (2.7%)	19 (8.2%)	
**Family members (persons/household)**				
Average household members = 5 people/household				0.064
**State of residential occupancy**				
Owner occupied	54 (11.9%)	30 (13.6%)	24 (10.3%)	0.154
Squatter	216 (47.7%)	110 (50%)	106 (45.5%)	
Tenant	106 (23.4%)	42 (19.1%)	64 (27.5%)	
Living with a host family	21 (4.6%)	13 (5.9%)	8 (3.4%)	
Others	56 (12.4%)	25 (11.4%)	31 (13.3%)	
**Highest Educational level**				
None	79 (17.4%)	19 (8.6%)	60 (25.8%)	<0.001 *
Primary	92 (20.3%)	30 (21.5%)	62 (26.6%)	
Secondary	112 (24.7%)	54 (22.7%)	58 (24.4%)	
Tertiary	145 (32%)	89 (40.5%)	56 (22%)	
Others	25 (5.6%)	20 (6.7%)	5 (1.2%)	
**Occupation**				
Trader	65 (14.3%)	35 (15.9%)	30 (12.9%)	<0.001 *
Daily wage-earner	153 (33.8%)	81(36.8%)	72 (30.9%)	
Public Servant	1 (0.2%)	1 (0.4%)	-	
Unemployed	101 (22.3%)	40 (18.6%)	61 (25.8%)	
Student	81 (17.9%)	17 (7.7%)	64 (27.5%)	
Private employee	23 (5.1%)	23 (10.3%)	-	
Others	29 (6.4%)	23 (10.3%)	6 (2.9%)	

^a^ Fisher’s Exact test, * Significance level ≤ 0.05.

**Table 2 healthcare-10-01097-t002:** Awareness and practices of COVID-19 protection.

Variable	Frequency *n* = 453 (%)	Non-Access to Digital Information (%)	Access to Digital Information (%)	Chi-Square	df	*p*-Value
**Types of COVID-19 WASH information**						
How to protect yourself	400 (72%)	209 (89.7%)	191 (86.8%)	0.90	1	0.340
What to do in case of infection	213 (38%)	104 (44.6%)	109 (49.5%)	1.095	1	0.295
Government response measure	133 (24%)	60 (25.8%)	73 (33.2%)	3.013	1	0.083
How to protect elderly/vulnerable	135 (30%)	67 (28.8%)	68 (30.9%)	0.254	1	0.616
How to behave in the public	202 (36%)	103 (44.4%)	99 (45.0%)	0.017	1	0.897
COVID situation reports in Thailand (Number of Infection Cases, Death, Recovered)	302 (55%)	142 (60.9%)	160 (72.7%)	7.070	1	0.008 *
COVID global case and situation reports	151 (27%)	67 (28.8%)	84 (38.2%)	4.525	1	0.033 *
Urgent announcement/notice/measure from the government (e.g., Lockdown area, State quarantine)	97 (17.5%)	37 (15.9%)	60 (27.3%)	8.592	1	0.003 *
**Sources of COVID-19 WASH information**						
Posters	23 (4%)	11 (5.0%)	12 (5.2%)	0.005	1	0.942
Local television	359 (65%)	179 (76.8%)	180 (81.8%)	1.716	1	0.190
Government COVID-19 websites	20 (4.4%)	2 (1.2%)	18 (6.0%)	1.566	1	0.211
Neighbors/friends	79 (14%)	38 (16.3%)	41 (18.6%)	0.426	1	0.514
Newspapers	13 (2.9%)	6 (2.6%)	7 (3.2%)			0.783 ^a^
Radio	18 (3%)	6 (2.6%)	12 (5.5%)			0.150 ^a^
Others	7 (1%)	2 (0.9%)	5 (2.3%)			0.273 ^a^

* Significance level ≤ 0.05. ^a^ Fisher’s Exact test.

**Table 3 healthcare-10-01097-t003:** Factors influencing urban marginalized people’s access to COVID-19 digital information.

Variable	Access to Online Information	
	Beta	*p*-Value
Age	−0.610	0.000 *
Gender	−0.110	0.003 *
Nationality	−0.169	0.000 *
Legal marital status	−0.037	0.369
Total family members	−0.037	0.321
State of residential occupancy	−0.059	0.118
Highest educational level	0.012	0.783
Occupation	0.032	0.409
Model *p*-value	<0.001 *	
R^2^	0.411	

* Significance level ≤ 0.05.

## Data Availability

Not applicable.

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
