# Peer review of "Access to Digital Information and Protective Awareness and Practices towards COVID-19 in Urban Marginalized Communities"

_healthcare, 2022, doi:10.3390/healthcare10061097_

Round 1

Reviewer 1 Report

The manuscript showed a vital role of digital communication during COVID-19 pandemic. Based on their results, although some people cannot access to digital information, the practice of COVID-19 protection is similar, fortunately. The reason may be due to the family size, teenagers and elderly people can get information from their family members. The study did not include the indirect access to digital information, which can be mentioned a little bit in the discussion.

Major revision:

Table 1, p-value is not clear. The authors showed p-value in each category such as age, gender, etc. However, in each category, there should be different p-value for each sub-category. For example, in the Age, p-value of group 18-35 and 36-59 should be different. How the authors calculate p-values? In addition, Table 1 table legend did not indicate superscribe "a" and "*"

What is sample size for Figure 2? The authored showed 220 people have access and 233 without access, most likely half and half. However, Figure 2 showed 18% people don’t use anything, plus 1% no internet mobile phone. The percentage does not match. It would be better for the authors to clarify which one is digital information. In addition, the total percentage for Figure 2 is 87%, not 100%.

What is the total percentage of each category in Figure 4? Percentage of respondents? Figure 4 figure legend did not show the light orange and the orange ones.

Minor revision:

Page 5 format need to be revised.

Author Response

Dear Reviewer,

Thank you very much for your constructive and helpful comments that helped us improve the quality of our manuscript. We have revised the manuscript and highlighted it in red color and carefully responded to your one by one comment as follows:

Major revision:

1. p-values reported in Table 1 are the result of Fisher’s Exact Test. We aimed to use Fisher’s Exact Test to determine whether or not there is a significant difference between two or among more categorical variables (number of rows) in two sample groups (number of columns) of access to digital communication and non-access to digital communication.

2. The sample size in Figure 2 is n= 220. Data in this Figure 2 is now verified and corrected. The total sample size has been also added in the figure title.

3. The total percentage of each category in Figure 4 is 100% of 453 respondents, including “No Idea” responses. The figure has been ow revised to be consistent,

Minor revision:

4. Page 5 has been now re-formatted as suggested 

Reviewer 2 Report

The topic is very interesting and topical. It requires elaboration:

1. I encourage you to work out hypotheses. The authors formulated the questions of the researchers. There are no hypotheses.

2. Please describe the methods adopted for the analysis of the survey.

3. Please indicate the limitations in the implementation of the research.

4. Please indicate the scientific and cognitive implications of this research.

Author Response

Dear Reviewer,

Thank you very much for your appreciation of our research and your constructive suggestions. We have revised the manuscript in accordance to your comments and responded to one by one comments as follows:

  1. I encourage you to work out hypotheses. The authors formulated the questions of the researchers. There are no hypotheses.

The hypothesis is now added between Line 60-62

  1. Please describe the methods adopted for the analysis of the survey.

We believe that the data analysis methods of the survey have been described clearly in 2.3 Data analysis

  1. Please indicate the limitations in the implementation of the research.

Limitation in the implementation of the research is now added in Line 228-230

  1. Please indicate the scientific and cognitive implications of this research.

The scientific and cognitive implications of this research are reported in Conclusion between Lines 236-245

Reviewer 3 Report

In this manuscript, the authors have investigated how access to digital information influences the vulnerabilities of urban marginalized communities to the COVID-19 pandemic. Data was collected by questionnaire in two slum communities of Bangkok city, Thailand. The study has highlighted how access to digital information is beneficial to the whole community for responding to and minimizing the risks of COVID-19.

I suggest that this manuscript could be suitable for publication in Healthcare if the text is subjected to a substantial number of minor revisions, as follows:

Line 5.  Insert a space after “&”

Lines 9-10.  Remove “Department of Development and Sustainability, School of Environment, Resources and Development, Asian Institute of Technology, Pathum Thani 12120, Thailand” – this affiliation has been repeated

Line 20.  Change “the study” to “this study”

Line 22.  Change “Covid-19” to “COVID-19”

Line 23.  Change “Covid-19” to “COVID-19” and insert “A total of” before “453”

Line 27.  Change “Covid-19” to “COVID-19”

Line 29.  Change “Covid-19” to “COVID-19”

Line 30.  Change “Covid-19” to “COVID-19”

Line 38.  Insert “COVID-19” after “coronavirus”

Line 46.  Insert a full-stop after “[7]”

Lines 46-47.  Change “Covid-19” to “COVID-19”

Line 50.  Change “Covid-19” to “COVID-19”

Line 54.  Change “Covid-19” to “COVID-19”

Line 57.  Insert “are the” before “levels”

Line 58.  Change “Covid-19” to “COVID-19”

Line 63.  Change “was selected purposely as the” to “were selected for the”

Line 64.  Insert a full-stop after “[13]”

Line 65.  Change “They” to “There”

Line 67.  Change “communities such as” to “communities, including”

Line 68.  Change “condition caused them hard to” to “conditions, this has made it difficult to” and “services as well as there are no inadequate” to “services, and there are no adequate”

Line 71.  Change “was designed by using” to “used”

Lines 73-74.  Change “in two subcommunities including Lock 1-2-3 sub-communities and Ban Guay sub-communities with a total population is 9,125 populations” to “in the two subcommunities of Lock 1-2-3 and Ban Guay, with a total population is 9,125”

Line 78.  Is this formula complete?

Line 83.  Change “when surveyed due to this method is suitable” to “as this is suitable”

Line 86.  Insert “with” after “233”

Line 88.  Change “have been” to “were”

Line 92.  Change “ethic” to “ethnicity”

Line 94.  Change “occupy” to “occupancy”

Line 95.  Insert a full-stop after “sources”

Line 96.  Insert a full-stop after “practices”

Line 97.  Change “in the Thai” to “using Thai”

Line 98.  Change “intensive” to “extensive”

Line 101.  Change “Lock 1-2-3 sub-communities and Ban Guay” to “Lock 1-2-3 and Ban Guay”

Line 104.  Change “Statistic” to “Statistical”

Line 105.  Change “Chi-square” to “The Chi-square test”

Line 107.  Change “significant” to “significance”

Line 111.  Change “ethics” to “ethnicity”

Lines 121-122.  Remove “This section may be divided into subheadings. It should provide a concise and precise description of the experimental results, their interpretation, as well as the experimental conclusions that can be drawn.”

Line 123.  Change “Demographic” to “demographic” and “Respondents” to “respondents”

Line 125.  Change “showed” to “shows”

Line 127.  Change “are” to “were”

Lines 137-150.  The heading for section 3.1.2 and the beginning of the section have been mixed with Table 1

Line 153.  Change “figure” to “Figure”, this line refers to Figure 3, what about Figure 2?

Line 155.  Change “helping” to “help”

Line 156.  Change “daily received” to “received daily”

Line 164.  Change “Covid-19” to “COVID-19”

Line 165.  Change “Covid-19” to “COVID-19”

Line 167.  Change “Covid-19” to “COVID-19”

Line 168.  Define “NGO”

Line 172.  Change “showed” to “shows”

Lines 173-174.  Rewrite “to have higher perceived about ‘COVID situation in Thailand’ (x2 = 7.070, df = 1, p=0.008), the global situation”

Lines 174-175.  Rewrite “had perceived more the urgent announcements and news”

Line 184.  Change “Covid-19” to “COVID-19”

Line 205.  Change “website” to “websites”

Line 206.  Remove “the”

Line 211.  Rewrite “better health protection actions and protective intentions than”

Line 214.  Change “empathized” to “emphasized”

Line 216.  Change “Covid-19” to “COVID-19”

Line 217.  Change “toward” to “towards”

Line 220.  Remove “the”

Lines 271-356.  Please check the required format for the reference list

Lines 275-276.  Reference 3 is not complete.

Author Response

Dear Reviewer,

Thank you very much for your helpful comments that helped us improve the quality of our paper. We have now revised all errors of grammar and spelling as suggested:

Line 5.  Insert a space after “&”.  Revised

Lines 9-10.  Remove “Department of Development and Sustainability, School of Environment, Resources and Development, Asian Institute of Technology, Pathum Thani 12120, Thailand” – this affiliation has been repeated.  Revised

Line 20.  Change “the study” to “this study”. Revised

Line 22.  Change “Covid-19” to “COVID-19”. Revised

Line 23.  Change “Covid-19” to “COVID-19” and insert “A total of” before “453”. Revised

Line 27.  Change “Covid-19” to “COVID-19”. Revised

Line 29.  Change “Covid-19” to “COVID-19”. Revised

Line 30.  Change “Covid-19” to “COVID-19”. Revised

Line 38.  Insert “COVID-19” after “coronavirus”. Revised

Line 46.  Insert a full-stop after “[7]”. Revised

Lines 46-47.  Change “Covid-19” to “COVID-19”. Revised

Line 50.  Change “Covid-19” to “COVID-19”. Revised

Line 54.  Change “Covid-19” to “COVID-19”. Revised

Line 57.  Insert “are the” before “levels”. Revised

Line 58.  Change “Covid-19” to “COVID-19”. Revised

Line 63.  Change “was selected purposely as the” to “were selected for the”. Revised

Line 64.  Insert a full-stop after “[13]”. Revised

Line 65.  Change “They” to “There” . Revised

Line 67.  Change “communities such as” to “communities, including”. Revised

Line 68.  Change “condition caused them hard to” to “conditions, this has made it difficult to” and “services as well as there are no inadequate” to “services, and there are no adequate”. Revised

Line 71.  Change “was designed by using” to “used”. Revised

Lines 73-74.  Change “in two subcommunities including Lock 1-2-3 sub-communities and Ban Guay sub-communities with a total population is 9,125 populations” to “in the two subcommunities of Lock 1-2-3 and Ban Guay, with a total population is 9,125”. Revised

Line 78.  Is this formula complete? – Word error causes formula lost, I fixed it. Revised

Line 83.  Change “when surveyed due to this method is suitable” to “as this is suitable”. Revised

Line 86.  Insert “with” after “233”. Revised

Line 88.  Change “have been” to “were”. Revised

Line 92.  Change “ethic” to “ethnicity”. Revised

Line 94.  Change “occupy” to “occupancy”. Revised

Line 95.  Insert a full-stop after “sources”. Revised

Line 96.  Insert a full-stop after “practices”. Revised

Line 97.  Change “in the Thai” to “using Thai”. Revised

Line 98.  Change “intensive” to “extensive”. Revised

Line 101.  Change “Lock 1-2-3 sub-communities and Ban Guay” to “Lock 1-2-3 and Ban Guay”. Revised

Line 104.  Change “Statistic” to “Statistical”. Revised

Line 105.  Change “Chi-square” to “The Chi-square test”. Revised

Line 107.  Change “significant” to “significance”. Revised

Line 111.  Change “ethics” to “ethnicity”. Revised

Lines 121-122.  Remove “This section may be divided into subheadings. It should provide a concise and precise description of the experimental results, their interpretation, as well as the experimental conclusions that can be drawn.”. Revised

Line 123.  Change “Demographic” to “demographic” and “Respondents” to “respondents”. Revised

Line 125.  Change “showed” to “shows”. Revised

Line 127.  Change “are” to “were”. Revised

Lines 137-150.  The heading for section 3.1.2 and the beginning of the section have been mixed with Table 1. Revised

Line 153.  Change “figure” to “Figure”, this line refers to Figure 3, what about Figure 2? Revised

Line 155.  Change “helping” to “help”. Revised

Line 156.  Change “daily received” to “received daily”. Revised

Line 164.  Change “Covid-19” to “COVID-19”. Revised

Line 165.  Change “Covid-19” to “COVID-19”. Revised

Line 167.  Change “Covid-19” to “COVID-19”. Revised

Line 168.  Define “NGO”. Revised

Line 172.  Change “showed” to “shows”. Revised

Lines 173-174.  Rewrite “to have higher perceived about ‘COVID situation in Thailand’ (x2 = 7.070, df = 1, p=0.008), the global situation”. Revised

Lines 174-175.  Rewrite “had perceived more the urgent announcements and news”. Revised

Line 184.  Change “Covid-19” to “COVID-19”. Revised

Line 205.  Change “website” to “websites”. Revised

Line 206.  Remove “the”. Revised

Line 211.  Rewrite “better health protection actions and protective intentions than” revised

Line 214.  Change “empathized” to “emphasized”. Revised

Line 216.  Change “Covid-19” to “COVID-19”. Revised

Line 217.  Change “toward” to “towards”. Revised

Line 220.  Remove “the”. Revised

Lines 271-356.  Please check the required format for the reference list revised

Lines 275-276.  Reference 3 is not complete. Revised and completed